# SARS-CoV-2 Variants in Lebanon: Evolution and Current Situation

**DOI:** 10.3390/biology10060531

**Published:** 2021-06-14

**Authors:** Nancy Fayad, Walid Abi Habib, Ahmed Kandeil, Rabeh El-Shesheny, Mina Nabil Kamel, Youmna Mourad, Jacques Mokhbat, Ghazi Kayali, Jimi Goldstein, Jad Abdallah

**Affiliations:** 1School of Pharmacy, Lebanese American University, P.O. Box 36, Byblos, Lebanon; nancy.fayad@lau.edu.lb (N.F.); walid.abihabib01@lau.edu.lb (W.A.H.); 2Center of Scientific Excellence for Influenza Viruses, National Research Centre, Giza 12622, Egypt; ahmed.kandeil@human-link.org (A.K.); rabeh.elshesheny@stjude.org (R.E.-S.); minanabil56@yahoo.com (M.N.K.); 3St. Jude Children’s Research Hospital, 262 Danny Thomas Place, Memphis, TN 38105, USA; 4Al Hadi Laboratory and IVF Center, P.O. Box 44, Beirut, Lebanon; youmna_mourad@hotmail.com; 5School of Medicine, Lebanese American University, P.O. Box 36, Byblos, Lebanon; jacques.mokhbat@lau.edu.lb; 6Human Link, Dubai 971, United Arab Emirates; ghazi@human-link.org; 7Department of Epidemiology, Human Genetics, and Environmental Sciences, University of Texas, Houston, TX 77030, USA; 8School of Engineering and Technology, University of Hertfordshire, Hatfield, Hertfordshire AL10 9AB, UK

**Keywords:** COVID-19, Lebanon, GISAID, clades, mutations, phylogenetic relationship

## Abstract

**Simple Summary:**

Thanks to whole genome sequencing approaches, the proteins that compose the SARS-CoV-2 virus that causes COVID-19 are compared, and mutations and differences highlighted. This allows strains to be classified into clades or lineages, according to marker mutations in their proteins as well as relatedness to certain epidemiological events. In this study, 58 SARS-CoV-2 Lebanese strains were analyzed. They were classified into four GISAID clades and 11 Pango lineages. Moreover, the mutational survey revealed several mutations in the encoded proteins, particularly the structural ones, in which 22 uncommon mutations were found, 21 of which were in strains sequenced within this study. The latter also showed an interesting combination of mutations in their spike protein, the key element in the virus’ interaction with cellular receptors. In summary, this study highlights the key features of sequenced Lebanese SARS-CoV-2 genomes, including their classification, phylogenetic relationship, and mutations.

**Abstract:**

The severe acute respiratory syndrome coronavirus 2 (SARS-CoV-2) has seen a worldwide spread since its emergence in 2019, including to Lebanon, where 534,968 confirmed cases (8% of the population) and 7569 deaths have been reported as of 14 May 2021. With the genome sequencing of strains from various countries, several classification systems were established via genome comparison. For instance, the GISAID clades classification highlights key mutations in the encoded proteins that could potentially affect the virus’ infectivity and transmission rates. In this study, 58 genomes of Lebanese SARS-CoV-2 strains were analyzed, 28 of which were sequenced for this study, and 30 retrieved from the GISAID and GenBank databases. We aimed to classify these strains, establish their phylogenetic relationships, and extract the mutations causing amino acid substitutions within, particularly, the structural proteins. The sequenced Lebanese SARS-COV-2 strains were classified into four GISAID clades and 11 Pango lineages. Moreover, 21 uncommon mutations in the structural proteins were found in the newly sequenced strains, underlining interesting combinations of mutations in the spike proteins. Hence, this study constitutes an observation and description of the current SARS-CoV-2 genetic and clade situation in Lebanon according to the available sequenced strains.

## 1. Introduction

The severe acute respiratory syndrome coronavirus 2 (SARS-CoV-2), the causal agent of coronavirus disease (COVID-19), emerged in the Wuhan district, China, in December 2019 [1]. Since then, this virus has spread around the globe and, on 11 March 2020, was officially declared a global pandemic by the World Health Organization (WHO) [2]. Globally, several strategies were adopted to fight SARS-CoV-2-induced alveolar damages and reduce the symptoms’ severity. A strategy based on combinational drug therapy involving antimalarial, antifilarial, and antiviral drugs was adopted in several countries [3]. Nonetheless, the recent development of several vaccines and the ongoing inoculation of the population constitutes a hope for an end to this pandemic.

SARS-CoV-2 is an enveloped virus, belonging to the *Coronaviridae* family, with a nonsegmented, positive-sense, single-stranded RNA. Its genome holds 5′ and 3′ untranslated regions (UTR), in addition to ten open reading frames (ORF) encoding structural, nonstructural, and accessory proteins [4]. ORF1ab is proteolytically cleaved into 16 nonstructural proteins (NSP1-16), whereas ORFs 2, 4, 5, and 9 encode the four structural proteins S (spike), E (envelope), M (membrane), and N (nucleocapsid), respectively [5,6]. In general, RNA viruses are prone to a high mutation rate, up to a million times that of their host cells. This property is key to their survival and evolution, where some mutations can affect the virus’ transmission and/or pathogenicity [7].

Similar to other RNA viruses, more than 10,000 single nucleotide polymorphisms (SNPs) were reported for SARS-CoV-2 in only a few months, following the comparison of its genomes [8,9]. This number is increased by (I) the naturally high mutation rate, (II) community transmission, and (III) global transmission. These SNPs and their effect on the encoded proteins lead to the emergence of new SARS-CoV-2 variants and the subsequent establishment of SARS-CoV-2 classification systems. The Global Initiative on Sharing All Influenza Data (GISAID, [10]) introduced a nomenclature structure, which encompasses eight phylogenetic clades based on marker mutations. Clade “L”, which holds the original SARS-CoV-2 strain, Wuhan-Hu-1 (Refseq accession number: NC_045512.2; [11]), forms alongside clade “S”, the two major clades that emerged at the start of the pandemic in 2019. A subsequent division of clade “L” into “V” and “G”, was brought forth by the marker mutation L37F in NS6 and G251V in NS3 for clade “V”, and mutation D614G in S for clade “G”. Then, “G” was split into “GV”, “GH”, and “GR”, due to the mutations NS3 W57H, S A222V, and N G204R, respectively. Clade “GRY” then emerged from clade “GR” in December 2020, because of a combination of ten marker mutations in S. To these eight clades, an “O” clade was added, which is a generic, less clearly defined group [12,13,14,15].

Amino acid changes, particularly in the structural proteins, can affect the infectivity of the strain as well as the severity of the symptoms. For instance, a recent study in Singapore concluded that, on the one hand, strains belonging to clades L and V showed an increased production of proinflammatory cytokines, hence a more pronounced systemic inflammatory reaction in the lungs and upper airways [16]. On the other hand, strains belonging to the clade G or one of its derivates, i.e., presenting the D614G mutation in S, were shown to have higher infectivity and increased viral loads. This is likely due to a change in the conformation of the S protein, resulting in better binding with the host’s receptor protein, the angiotensin-converting 2 (ACE2) protein [17,18,19].

The GISAID clade classification provides a broad overview, in contrast to a more detailed phylogenetic assignment, the Pango lineage, which identifies a cluster according to the sequence relatedness and the association with an epidemiological event [20]. This dynamic nomenclature system uses a phylogenetic framework to classify and track lineages that contribute to active spread, while simultaneously flagging those that are no longer observed and are probably inactive. Two primary lineages, A and B, were assigned based on the divergence of the few first SARS-CoV-2 strains. Then, further specifications were given (e.g., A.2 or B.1) based on ancestor/descendant relationship between strains. When a maximum of three sublevels is reached, a new letter is assigned to indicate that a new level has emerged. In addition to being indicative of the phylogenetic relationships between the strains, this classification allows a reversion to the country of origin [20].

Lebanon confirmed its first case of COVID-19 on 21 February 2020, after a 45-year-old woman returning from Iran tested positive [21]. In the following months, the number of cases increased but remained limited because of strict national confinement, closed borders, and precautionary measures. Then, the country witnessed a surge in the number of confirmed COVID-19 positive cases due to the deteriorating socioeconomic situation and the Beirut port explosion that occurred on 4 August 2020, among other aspects, to reach 534,968 positive cases and 7569 deaths on 14 May 2021 [21,22,23,24,25].

The genomes of 11 Lebanese strains were previously analyzed and showed a number of mutations in the ORF of structural and nonstructural proteins [26,27]. A total of 40 single nucleotide variations were detected in the analyzed strains and resulted in amino acid changes, a stop codon, a frameshift, and a deletion. These variations were found in the ORF1ab, S, NS3 (ORF3a), E, ORF7a, and N proteins. Mutations S D614G and NS3 Q57H were the most frequent among these strains, found in 7 and 6 of the 11 strains, respectively [26,27].

We analyzed the genome of 58 Lebanese circulating strains collected between February 2020 and January 2021, out of which 28 were sequenced for this study and 30 were previously published sequences. Consequently, we generated the phylogenetic relationships between the Lebanese strains and strains from around the world. Moreover, we studied the mutations present in these 58 strains, particularly the mutations located in the four structural proteins S, N, M, and E. This study thus allowed us to highlight the SARS-CoV-2 variants circulating within the Lebanese population.

## 2. Materials and Methods

### 2.1. Sample Collection

Nasopharyngeal and oropharyngeal swabs of confirmed COVID-19 patients were collected from the Lebanese American University (LAU) Medical Center-Rizk Hospital and Al-Hadi Medical Center. Twenty-eight samples were collected between 18 March and 1 December 2020 and labeled with the prefix LAU-R (24 strains) and LB-R (4 strains).

### 2.2. Sample Processing and SARS-CoV-2 Genome Sequencing

The total SARS-CoV-2 nucleic acid was extracted from 140 μL of each swab sample using a QIAmp viral RNA mini kit (cat. No. 52906; QIAGEN, Düsseldorf, Germany) according to the manufacturer’s instructions. Extracted viral RNA was stored at −80 °C until further use. The presence of SARS-CoV-2 was confirmed by real-time PCR using a Verso 1-step RT-qPCR Kit (cat. No. AB4100C; Thermo Fisher Scientific, Massachusetts, USA), primer F-S COV2-NRC (5′-TACCCATTGGTGCAGGTATATGC-3′), and primer R-S COV2-NRC (5′-GTGAGGCAATGATGGATTGACTA-3′) targeting the S and ORF genes in a 40 cycle run. Complementary DNA (cDNA) was generated using the SuperScript™ IV One-Step RT-PCR kit (cat. No. 12594025; Thermo Fisher Scientific, Massachusetts, USA) for genome sequencing, using the following protocol: reverse transcription for 10 min at 50 °C; initial PCR activation for 2 min at 98 °C, followed by standard initial denaturation for 10 sec at 98 °C; annealing for 10 sec at 42 °C; extension for 1 min at 72 °C, with repeating the last three steps 40 times; and then final extension for 5 min at 72 °C. Next generation sequencing (NGS) of the SARS-CoV-2 genome was then performed using a Nextera Illumina protocol as per manufacturer instructions at the Multi-Omics Laboratory at the Lebanese American University (for LAU-R strains) and Saint Jude Children’s Research Hospital, USA, for (LB-R strains).

### 2.3. Bioinformatic Analysis

The genome sequences of 30 previously published SARS-CoV-2 Lebanese strains were recovered from GISAID and GenBank, in addition to the 28 strains sequenced in this study. Their collection date was from 21 February 2020 to 8 January 2021. Those 58 genomes were assigned a clade and a Pango lineage based on marker mutations, via the built-in GISAID detector and the Pangolin COVID-19 Lineage Assigner (https://pangolin.cog-uk.io/, last accessed on 5 May 2021). Five randomly chosen “reference” strains from each of the eight clades designated by GISAID and the NCBI Refseq Wuhan-Hu-1 strain (Refseq accession number: NC_045512.2) were included in the phylogenetic analysis. In total, 99 sequences were aligned using the MUSCLE algorithm on MegaX (version 10.0.5; [28]). A phylogenetic tree was then constructed via a maximum likelihood approach with a bootstrap value of 500. The resulting tree was rooted with the branch holding the Wuhan-Hu-1 strain.

Furthermore, the mutations present in the 58 SARS-CoV-2 genomes of the Lebanese isolates were analyzed using “CoVsurver: The Mutation Analysis of hCoV-19” (https://www.gisaid.org/epiflu-applications/covsurver-mutations-app/; last accessed on 5 May 2021). The amino acid sequences of the translated ORF were compared to those of the reference strain Wuhan_WIV04 (GISAID accession number: EPI_ISL_402124; GISAID classification Clade: L; Pango Lineage: B). All mutations present in any of the viral protein were examined, particularly those present in the structural proteins S, N, M, and E. Mutations caused by the presence of “N”, hence resulting in an “X” amino acid, were disregarded. Then, detected amino acid substitutions that are considered uncommon according to CoVsurver were cross checked with NCBI’s “Mutations in SARS-CoV-2 SRA Data” (https://www.ncbi.nlm.nih.gov/labs/virus/vssi/#/scov2_snp; last accessed on 5 May 2021), which compares SARS-CoV-2 sequences to the NCBI reference strain Wuhan-Hu-1 (Refseq accession number: NC_045512.2).

## 3. Results

### 3.1. SARS-CoV-2 Lebanese Strains: Sequences and Classification

Fifty-eight Lebanese SARS-CoV-2 strains were sequenced and submitted to the GISAID and NCBI GenBank depositories, out of which 28 were sequenced within this study. The genome length of the latter strains ranged from 28,846 to 29,899 bp. As for the other 30 Lebanese strains, their genome was recovered from the GISAID and GenBank depositories, and its length ranged from 29,802 to 29,851 bp. In comparison to the GISAID reference sequence Wuhan_WIV04 (GISAID accession number: EPI_ISL_402124), the insertion or deletion of one or more nucleotides was seen in the genome of several strains (Appendix A). A frequently occurring deletion, detected in nine strains, consisted of a single nucleotide at position 9850 (position in the reference genome) within the ORF1ab gene, more precisely at the NS4, although no change was detected in direct link to this deletion. In other cases, the insertion of one or more nucleotides was also reported in comparison with Wuhan_WIV04, the longest one being an insertion of 12 nucleotides in strain LB-R12, but that also resulted in no change in the protein sequences (Appendix A).

Each strain was placed in a GISAID clade, a classification that allows grouping of strains based on marker mutations. For the analyzed strains, clade G included 48% of the strains (28 out of 58 strains). Strains associated with this clade were collected between 19 March 2020 and 1 December 2020. Clades GH and GRY had 21% (12 out of 58) and 17% (10 out of 58), respectively (Figure 1A).

A time course visualization of clade distribution of the 58 analyzed strains showed that clades G and GH overlapped and were the most prevalent in early and mid-2020. However, this was no longer the case in December 2020–January 2021, in which clade GRY took over (Figure 2). This corresponds to what was seen in the time course visualization for the available sequences in the GISAID database ([29]; last updated 5 May 2021).

Moreover, no strains were detected from clades GV, L, V, and S, whereas three strains, S3, S5, and S9, were detected from clade O, a less clearly defined group. Strains S5 and S9 were analyzed in a previous study and identified as belonging to the Nextstrain clade 19A, a group associated with the GISAID clades L or V. As for strain S3, it was excluded in the mentioned study, due to a poor sequence quality [27].

The three most prevalent Pango lineages in the analyzed strains were B.1.398 (45%), B.1 (17%), and B.1.1.7 (17%) (Figure 1B). These lineages are associated with clades G, GH, and GRY, respectively. The Lebanese SARS-CoV-2 strains analyzed in this study and their collection date, GISAID and/or GenBank accession numbers, and assigned GISAID clades and Pango lineages are summarized in Table 1.

### 3.2. Phylogenetic Relationships

The genomes of the 58 Lebanese SARS-CoV-2 strains and 41 reference strains from each of the eight GISAID clades (Table 2) were aligned and their phylogenetic relationship assessed via a maximum likelihood approach. The resulting tree was rooted with the branch carrying the original SARS-CoV-2 strain Wuhan-Hu-1.

As expected, strains belonging to the same clade clustered together (Figure 3). Nevertheless, a phylogenetic distance was seen in some cases, such as for strains belonging to clade GH, where a small distance existed between strains LAU-R70, -R110, -R113, and -R170 and the rest of this clade’s strains (Figure 3). Another case is that of clade G, where the majority of the strains clustered together, except for the three strains LB-R8, LBN-9-B.1.258 Delta, and Germany BY chVir 929 (GISAID accession number: EPI_ISL_406862). The strains S3, S5, and S9 clustered with the clade V GISAID strains, suggesting that they may in fact belong to this clade, despite their previous classification in clade O, the less clearly defined GSIAID clade. Moreover, seven Lebanese SARS-CoV-2 strains clustered together on a separate branch, within the supercluster encompassing the clade G.

### 3.3. Mutation Survey: A Focus on Structural Proteins

The classification of the strains and the phylogenetic distance observed in some clades was reflected by the presence of mutations which resulted in a change of the amino acid composition of the encoded proteins. Hence, in order to further investigate the SARS-CoV-2 variants in Lebanon, a mutation survey of the Lebanese SARS-CoV-2 strains showed several amino acid substitutions in both nonstructural and structural proteins. The most commonly found amino acid mutation, detected in 13 out of the 58 strains, was P323L in NS12, the RNA-dependent RNA polymerase [30]. Another mutation, located in 12 strains, was the Q58H located in the accessory NS3, a transmembrane protein that forms ion channels in the host membrane [31].

As for the structural proteins, the number of amino acid changes within the Lebanese strains ranged from 1 to 14 in comparison to the GISAID reference strain Wuhan_WIV04 (Appendix A). The nine strains FAS/2021 and LBN_1 to 8 presented the highest number of amino acid changes in the structural proteins, whereas the eight strains LAU-R242, S1, S2 and S7–11 presented only one amino acid change in the structural proteins.

#### 3.3.1. Spike S Protein

For the S protein, 50 amino acid substitutions were detected (Figure 4a, Appendix A), with D614G being the most prevalent mutation within the analyzed strains (55 out of 58). This change is quite common and was previously reported in 96.73% of all GISAID Spike sequences in 170 countries. Furthermore, T95I was an amino acid change found in 19 of the Lebanese strains and was previously reported in 1.27% of all samples with S sequences in 79 countries. Moreover, T716I combined with nine other changes (S982A, D1118H, H69del, V70del, Y144del, N501Y, A580D, P681H, and D614G) marks the variant SARS-CoV-2 VUI 202012/0, also known as Pango lineage B.1.1.7 or clade GRY, which was first reported in the United Kingdom in December 2020 [32]. This combination of ten mutations was found in nine Lebanese strains collected between 24 December 2020 and 8 January 2021. Additionally, a tenth strain, Leb-UK-B, presented eight out of the ten substitutions, with D1118H and N501Y being absent.

While many of the detected mutations were known to be common, some having one million reports in GISAID at the time of the submission, others were less commonly found according to CoVsurver. Although some surveillance tools show the distribution of an amino acid substitution in accordance with when it was added to the database, the search for a substitution using CoVsurve was done based on the strain’s collection date, not the submission date of its genome. Various amino acid substitutions were detected once in the analyzed sequences and, according to GISAID’s CoVsurver, were either first or only reported in the corresponding strain (Table 3). Concerning the spike protein, a total of 15 amino acid changes were found in eight strains. While some were an “only report” by CoVsurver, this was in relation to the new amino acid, not the position of the change. This is the case of three S protein amino acid substitutions at positions G35, S31, and L5.

LAU-R186, collected on 1 August 2020, was one of the strains in which these uncommon mutations were found. In fact, LAU-R186 had ten amino acid substitutions in its S protein (Table 3; Appendix A). Three out of the ten changes are of particular interest: (I) N17S removes a potential N-glycosylation site at position 17 and has occurred 797 times in eight countries so far according to the GISAID’s CoVsurver, (II) F59Y has 106 reports until the time of the submission, and (III) Y38Stop can cause a 97.1% truncation of the protein. Another strain with an uncommon combination of amino acid changes is LAU-R45, collected on 1 June 2020. It has eight amino acid changes in its S protein, the most noteworthy being P39T, S31Y, and V11D (Table 3). S31Y is involved in the viral oligomerization interface, according to structural resolution of the S protein (PDB chain: 7cwm_B; data not shown). As for the substitution V11D, an equivalent mutation previously reported in the human MERS-CoV virus suggested an alteration in the antigenic drift [33]. The latter is also affected by a change in amino acid position L5, in which the change L5P was detected in strain LAU-R204, according to the analysis of the equivalent mutation L5F [18].

#### 3.3.2. Nucleocapsid N Protein

In total, 23 different amino acid substitutions were reported for the N protein in 36 of the 58 analyzed genomes, the most prevalent being R03K and G204R, both found in 15 strains (Figure 4b). These substitutions were commonly found in SARS-CoV-2 strains and registered over 500,000 occurrences in the GISAID database. Nevertheless, five amino acid substitutions in the N protein were uncommon (Table 3), four of which, at positions E367, L352, L407, and S318, were previously flagged for changes, but for a different substituting amino acid. The first was found to be involved in an alteration of antigenic drift, [34], whereas the last might be implicated in the viral oligomerization interface, according to the structural resolution of the N protein (PDB chain: 6wzq_D; data not shown). For example, L407F and L407L were previously reported, whereas L407M was labeled as an only report in the Lebanese LAU-R43 strain by GISAID’s CoVsurver and NCBI’s mutations data. Amino acid changes in positions 361–370 were reported in two of the analyzed strains. Moreover, the two stop mutations, E367Stop and L352Stop, respectively found in strains LAU-R145 and LAU-R231, may result in a 12.6 and 16.2% truncation of the protein.

#### 3.3.3. Membrane M and Envelope E proteins

Unlike S and N, the two remaining structural proteins presented only two amino acid substitutions, each detected in the Lebanese SARS-CoV-2 strains. For the M protein, the two amino acid changes were L16I, and A2V, previously reported 18 and 1866 times, respectively. As for the E protein, the two detected amino acid changes, C44S and V62D, were uncommon and detected in LAU-R110 and strain S6, respectively. The former might be implicated in the viral oligomerization interface, according to the structural resolution of the E protein (PDB chain: 5 × 30_A; data not shown).

## 4. Discussion

Lebanon, a small Mediterranean country with around 7 million inhabitants [35], presented more than 534,968 confirmed COVID-19 cases and 7569 deaths as of 14 May 2021 [25]. The current study forms an observational description of the 58 available genomes of Lebanese SARS-CoV-2 strains, their classification, phylogenetic relationships, and the mutations they hold. The Lebanese SARS-CoV-2 strains included the 28 sequenced for this study and 30 recovered from the GISAID and GenBank databases. Analysis was conducted on phylogenetic and mutational levels: each strain was assigned a GISAID clade and a Pango lineage, a phylogenetic analysis was done between the Lebanese strains and reference strains randomly chosen from the GISAID database, and the amino acid substitutions in the structural proteins were analyzed.

The classification according to the GISAID and Pango lineages revealed a prevalence of the clade G (48% of the strains) and its associated Pango lineage B.1.398 (45%). The main genetic marker of this clade was the amino acid substitution D614G, found in 94.8% (55 out of 58) of the analyzed strains. This substitution was shown to allow for an open conformational state of the S protein, thus enhancing the exposure of the receptor binding domain, which interacts with the ACE2 protein. This site is also the target for neutralizing antibodies. Moreover, this mutation was shown to increase the stability and infectivity rate of the virus in vitro [17]. This contributed to its rapid dissemination and its quick rise to dominant status, following its first appearance in January 2020, at the start of the pandemic [17]. The first confirmed COVID-19 incidence in Lebanon was on 21 February 2020, a time when the D614G mutation was already rapidly spreading, which explains its presence in 94.8% of the analyzed strains.

The clade GH, which differs from G by a marker mutation (NS3/ORF3a Q58H), included 21% of the strains along with its associated Pango lineage B.1 (17%). The clade GRY (18%) and its associated lineage B.1.1.7 included 17% of sequences. This clade was transmitted rapidly in December 2020 first through the UK, then Europe, then to more than 90 countries. It is considered a variant of concern by the Center for Disease Control and Prevention in the USA [36]. The series of amino acid changes in its S protein were behind an increase in transmission rates estimated to be 43 to 90% [37]. The phylogenetic analysis of the strains revealed a clustering mostly according to the GISAID clade and Pango lineage, with several subclades within each superclade, in addition to a number of phylogenetic distances that were noted between strains of the same clade. This might be explained by the subtle changes on the nucleotide level that do not necessarily translate into amino acid substitutions, but are taken into account on the DNA alignment level.

No strain belonged to clades GV, L, V, and S. For the first, it rapidly spread through Europe, mainly Italy, as of July 2020, reaching a peak in November 2020. However, according to the GISAID time course of clades distribution in collected sequences, the GV clade saw a steep decline as of January 2021 and is now being replaced by other clades [29]. As for clades L, V, and S, the first three clades designated at the beginning of the pandemic, detected in the GISAID sequence collection up to June 2020, their absence among the sequences of Lebanese SARS-CoV-2 strains could be explained by the fact that, up until June 2020, the community transmission of this virus in Lebanon was limited due to strict lockdown rules. The increase in the number of confirmed positive cases began in August 2020 [25], which coincides with a prevalence of clades G, GH, and GR, according to the GISAID database [29].

RNA viruses have a high mutation rate, a way to evolve and adapt to new hosts. In this study, we identified 77 amino acid substitutions in the structural proteins S, N, M, and E, 22 of which, located in LAU-R strains sequenced within this study, were labeled as uncommon (Table 3). Four of these substitutions resulted in a stop codon and the truncation of the corresponding protein in the range of 12 to 97%. However, mRNA in RNA viruses may be subject to a noncanonical translation, which includes a stop-codon readthrough, meaning that in a specific context, the resulting protein will be continued and not truncated at the site of the stop-codon [38,39].

The GISAID time course analysis of clade distribution in collected sequences shows that, as of late 2020, the GRY clade is taking over in lieu of G, GR, and GH, which were the dominant clades in mid-2020 [29]. When comparing this time course to the available SARS-CoV-2 Lebanese strains, we see that the G, GH, and GR could be gradually giving way to GRY in December 2020–January 2021.

Knowledge about the SARS-CoV-2 variants circulating among the Lebanese community is imperative and should be taken into account for further public health or epidemiological surveys and decisions. However, this study was limited by the small number of sequences analyzed. Additionally, among those sequences, the majority came from the same geographic area. This small number of sequences is due to the high cost of conducting full genome sequencing in Lebanon and complications associated with the political and economic situation of this country [22]. Hence, this may have affected the conclusions we are drawing in this study.

## 5. Conclusions

This study is the first to highlight 21 uncommon mutations in the structural proteins found in the newly sequenced LAU-R SARS-CoV-2 strains. Combined with other mutations found in these strains, LAU-R strains present an interesting combination of mutations in the spike protein. Moreover, this study was the first to classify 58 Lebanese SARS-CoV-2 strains in total, according to the GISAID clade classification and Pango lineage, and to compare this classification’s timeline with that of the GISAID sequence collection. Nevertheless, there is a need for a substantial improvement in the sequencing of Lebanese SARS-CoV-2 strains. Hence, future studies should follow a purposive sampling, bearing different sociodemographic statuses, such as a patient’s age, sex, and region of residence in addition to, most importantly, clinical presentation of disease.

## Figures and Tables

**Figure 1 biology-10-00531-f001:**
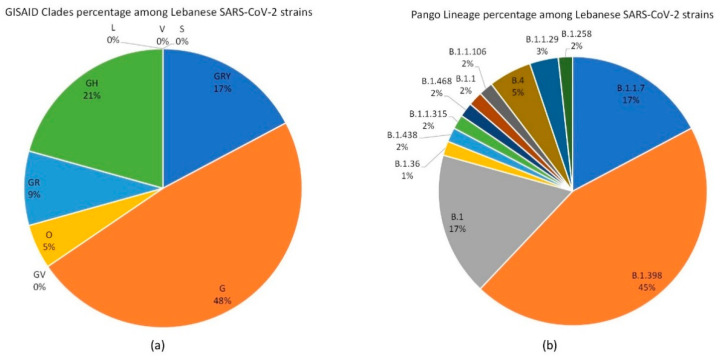
Percentage pie chart distribution of (**A**) GISAID clades and (**B**) Pango lineages among the 58 analyzed Lebanese SARS-CoV-2 strains. Each (**A**) clade or (**B**) lineage, as well as the percentages, are shown either on or just outside the corresponding section.

**Figure 2 biology-10-00531-f002:**
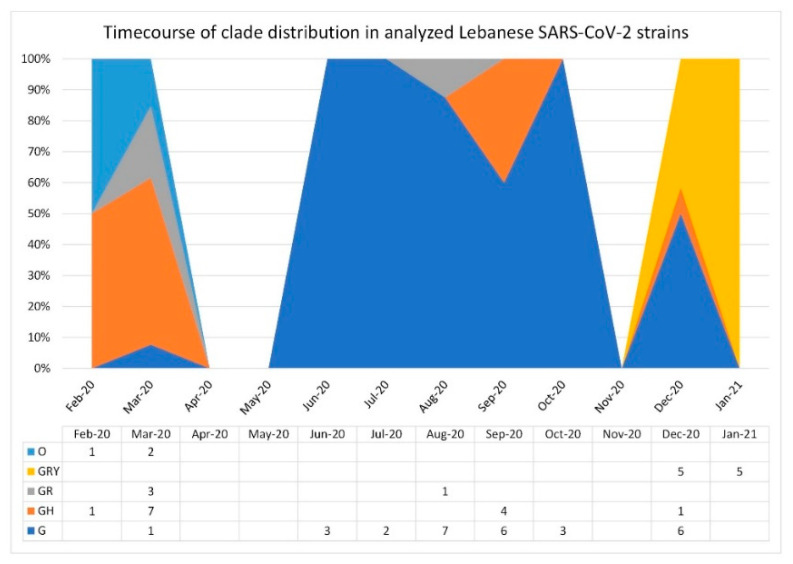
Time course GISAID clade distribution of the analyzed Lebanese SARS-CoV-2 strains over the last year. A data table with legend keys is shown directly under the chart.

**Figure 3 biology-10-00531-f003:**
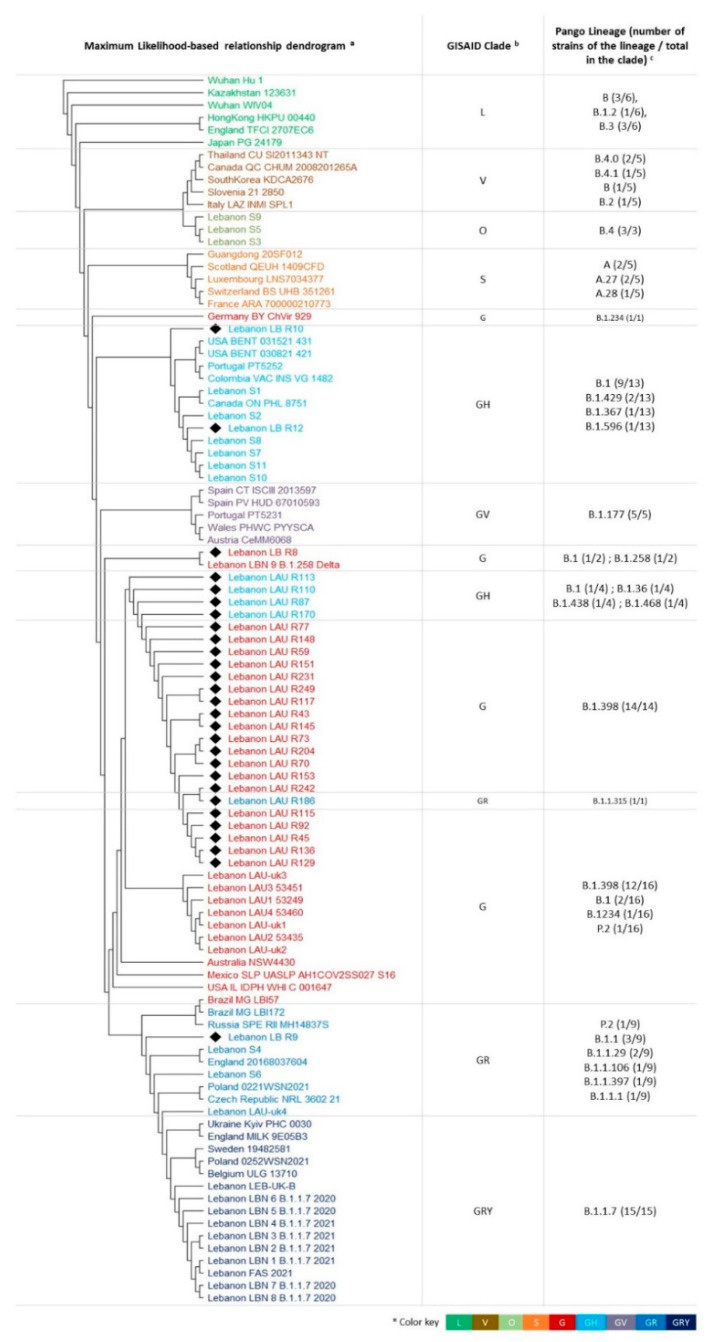
Phylogenetic tree representing the relationship among the 58 Lebanese SARS-CoV-2 and 41 reference genomes: (**a**) Using MegaX, genome sequences were first aligned with MUSCLE, and the relationship tree was established via maximum likelihood, with a bootstrap value of 500, followed by a rooting on Wuhan-Hu-1; (**b**) SARS-CoV-2 strains grouping according to the GISAID clades; (**c**) the various Pango Lineages within a certain cluster: the number of each lineage over the total number of strains within a cluster is shown. Color key is shown in the bottom right corner.

**Figure 4 biology-10-00531-f004:**
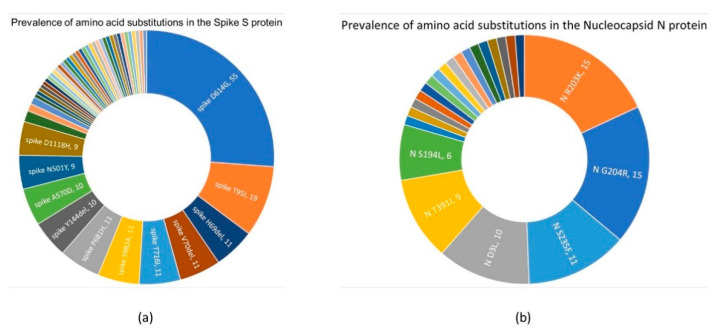
Various amino acid substitutions detected in: (**a**) the spike and (**b**) the nucleocapsid proteins. The number of strains in which each substitution was detected is shown next to said substitution on the wheel.

**Table 1 biology-10-00531-t001:** Analyzed Lebanese SARS-CoV-2 strains.

Virus Name	GISAID Accession Number (Genbank if Available)	Collection Date	GISAID Clade	Pango Lineage
FAS/2021	EPI_ISL_1159375	4-January-2021	GRY	B.1.1.7
LAU1-53249	EPI_ISL_637110	13-August-2020	G	B.1.398
LAU2-53435	EPI_ISL_637111	August-2020	G	B.1.398
LAU3-53451	EPI_ISL_637112	14-August-2020	G	B.1.398
LAU4-53460	EPI_ISL_637113	14-August-2020	G	B.1.398
LAU-R110 ^1^	EPI_ISL_1009085	1-September-2020	GH	B.1
LAU-R113 ^1^	EPI_ISL_1009123	1-September-2020	GH	B.1.36
LAU-R115 ^1^	EPI_ISL_1009127	1-September-2020	G	B.1.398
LAU-R117 ^1^	EPI_ISL_1009128	1-September-2020	G	B.1.398
LAU-R129 ^1^	EPI_ISL_1009159	1-September-2020	G	B.1.398
LAU-R136 ^1^	EPI_ISL_1009160	1-October-2020	G	B.1.398
LAU-R145 ^1^	EPI_ISL_1009213	1-October-2020	G	B.1.398
LAU-R148 ^1^	EPI_ISL_1009214	1-October-2020	G	B.1.398
LAU-R151 ^1^	EPI_ISL_1009617	1-September-2020	G	B.1.398
LAU-R153 ^1^	EPI_ISL_1009626	1-September-2020	G	B.1.398
LAU-R170 ^1^	EPI_ISL_1009628	1-September-2020	GH	B.1.438
LAU-R186 ^1^	EPI_ISL_1009629	1-August-2020	GR	B.1.1.315
LAU-R204 ^1^	EPI_ISL_1009630	1-July 2020	G	B.1.398
LAU-R231 ^1^	EPI_ISL_1009631	1-August-2020	G	B.1.398
LAU-R242 ^1^	EPI_ISL_1009656	1-December-2020	G	B.1.398
LAU-R249 ^1^	EPI_ISL_1009658	1-December-2020	G	B.1.398
LAU-R43 ^1^	EPI_ISL_982298	1-June-2020	G	B.1.398
LAU-R45 ^1^	EPI_ISL_1009011	1-June-2020	G	B.1.398
LAU-R59 ^1^	EPI_ISL_1009012	1-June-2020	G	B.1.398
LAU-R70 ^1^	EPI_ISL_1009013	1-July 2020	G	B.1.398
LAU-R73 ^1^	EPI_ISL_1009014	1-August-2020	G	B.1.398
LAU-R77 ^1^	EPI_ISL_1009015	1-August-2020	G	B.1.398
LAU-R87 ^1^	EPI_ISL_1009016	1-September-2020	GH	B.1.468
LAU-R92 ^1^	EPI_ISL_1009017	1-September-2020	G	B.1.398
LAU-uk1	EPI_ISL_768751	21-December-2020	G	B.1.398
LAU-uk2	EPI_ISL_768798	21-December-2020	G	B.1.398
LAU-uk3	EPI_ISL_768799	21-December-2020	G	B.1.398
LAU-uk4	EPI_ISL_768743	22-December-2020	GR	B.1.1
LB-R8 ^1^	EPI_ISL_498551 (MT801000)	19-March-2020	G	B.1
LB-R9 ^1^	EPI_ISL_498552 (MT801001)	18-March-2020	GR	B.1.1.106
LB-R10 ^1^	EPI_ISL_498554 (MT801002)	20-March-2020	GH	B.1
LB-R12 ^1^	EPI_ISL_498556 (MT801003)	20-March-2020	GH	B.1
LEB-UK-B	EPI_ISL_1072985	30-Decmember-2020	GRY	B.1.1.7
S1_758	EPI_ISL_450508	4-March-2020	GH	B.1
S2_759	EPI_ISL_450509	15-March-2020	GH	B.1
S3_760	EPI_ISL_450510	4-March-2020	O	B.4
S4_761	EPI_ISL_450511	4-March-2020	GR	B.1.1.29
S5_762	EPI_ISL_450512	27-February-2020	O	B.4
S6_766	EPI_ISL_450513	9-March-2020	GR	B.1.1.29
S7_763	EPI_ISL_454420	21-February-2020	GH	B.1
S8_767	EPI_ISL_450514	13-March-2020	GH	B.1
S9_764	EPI_ISL_450515	11-March-2020	O	B.4
S10_768	EPI_ISL_450516	15-March-2020	GH	B.1
S11_765	EPI_ISL_450517	9-March-2020	GH	B.1
LBN_1-B.1.1.7_2021	(MW686007)	4-January-2021	GRY	B.1.1.7
LBN_2-B.1.1.7_2021	(MW692113)	4-January-2021	GRY	B.1.1.7
LBN_3-B.1.1.7_2021	(MW692114)	8-January-2021	GRY	B.1.1.7
LBN_4-B.1.1.7_2021	(MW692115)	8-January-2021	GRY	B.1.1.7
LBN_5-B.1.1.7_2020	(MW692116)	28-December-2020	GRY	B.1.1.7
LBN_6-B.1.1.7_2020	(MW692117)	28-December-2020	GRY	B.1.1.7
LBN_7-B.1.1.7_2020	(MW692118)	24-December-2020	GRY	B.1.1.7
LBN_8-B.1.1.7_2020	(MW692119)	24-December-2020	GRY	B.1.1.7
LBN_9-B.1.258Delta	(MW720771)	12-December-2020	G	B.1.258

^1^ Strains sequenced in this study.

**Table 2 biology-10-00531-t002:** Randomly selected strains from each GISAID clade.

Virus Name ^1^	GISAID Accession Number (Genbank if Available)	Collection Date	GISAID Clade	Pango Lineage
England MLK-9E05B3	EPI_ISL_601443	20-September-2020	GRY	B.1.1.7
Ukraine Kyiv-PHC-0030	EPI_ISL_1495138	18-March-2021	GRY	B.1.1.7
Poland 0252WSN2021_wsseol	EPI_ISL_1493349	25-March-2021	GRY	B.1.1.7
Belgium ULG-13710	EPI_ISL_1493104	19-March-2021	GRY	B.1.1.7
Sweden 19482581	EPI_ISL_1492593	15-February-2021	GRY	B.1.1.7
England 20168037604	EPI_ISL_466615	16-February-2020	GR	B.1.1.1
Brazil MG-LBI172	EPI_ISL_1494976	22-February-2021	GR	P.2
Poland 0221WSN2021_wsseol	EPI_ISL_1493318	22-March-2021	GR	B.1.1
Czech Republic NRL-3602-21	EPI_ISL_1492191	10-February-2021	GR	B.1.1
Russia SPE-RII-MH14837S	EPI_ISL_1491732	9-February-2021	GR	B.1.1.397
Spain CT-ISCIII-2013597	EPI_ISL_539548	26-June-2020	GV	B.1.177
Spain PV-HUD-67010593	EPI_ISL_1495978	1-February-2021	GV	B.1.177
Austria CeMM6068	EPI_ISL_1495837	6-November-2020	GV	B.1.177
Portugal PT5231	EPI_ISL_1494894	9-February-2021	GV	B.1.177
Wales PHWC-PYYSCA	EPI_ISL_1476485	14-March-2021	GV	B.1.177
Canada ON-PHL-8751	EPI_ISL_418345	1-February-2020	GH	B.1.*
USA OR-TRACE-BENT-031521-431	EPI_ISL_1494938	14-March-2021	GH	B.1.429
Portugal PT5252	EPI_ISL_1494914	9-February-2021	GH	B.1.367
Colombia VAC-INS-VG-1482	EPI_ISL_1494952	5-March-2021	GH	B.1.596
USA OR-TRACE-BENT-030821-421	EPI_ISL_1494930	8-March-2021	GH	B.1.429
Wuhan WIV04 2019	EPI_ISL_402124	30-December-2019	L	B
England TFCI-2707EC6	EPI_ISL_1476828	3-April-2020	L	B.3
Japan PG-24179	EPI_ISL_1429761	19-February-2020	L	B.12
Kazakhstan 123631	EPI_ISL_1341150	1-March-2020	L	B
Hong Kong HKPU-00440	EPI_ISL_1289439	26-March-2020	L	B.3
Wuhan-Hu-1	(NC_045512.2)	1-December-2019	L	B
Italy LAZ-INMI-SPL1	EPI_ISL_412974	29-January-2020	V	B.2
South Korea KDCA2676	EPI_ISL_1490167	18-February-2020	V	B.41
Canada QC-CHUM-2008201265A	EPI_ISL_1378973	22-March-2020	V	B.40
Thailand CU-SI2011343-NT	EPI_ISL_1296450	2-April-2020	V	B.40
Slovenia 21-2850	EPI_ISL_1289874	30-March-2020	V	B
Guangdong 20SF012	EPI_ISL_403932	14-January-2021	S	A
France ARA-700000210773	EPI_ISL_1490236	11-March-2021	S	A.27
Scotland QEUH-1409CFD	EPI_ISL_1389947	18-March-2021	S	A
Switzerland BS-UHB-351261	EPI_ISL_1388124	25-January-2021	S	A.27
Luxembourg LNS7034377	EPI_ISL_1383855	1-February-2021	S	A.28
Germany BY-ChVir-929	EPI_ISL_406862	28-January-2020	G	B.1
Brazil MG-LBI57	EPI_ISL_1494965	27-January-2021	G	P.2
USA IL-IDPH-WHI-C-001647	EPI_ISL_1494503	14-July 2020	G	B.1.234
Australia NSW4430	EPI_ISL_1494721	3-April-2021	G	B.1
Mexico SLP-UASLP-AH1COV2SS027_S16	EPI_ISL_1494725	22-May-2020	G	B.1

^1^ The country of origin of each strain is indicated prior to its name.

**Table 3 biology-10-00531-t003:** List of uncommon AA substitutions detected in the structural proteins S, N, and E of the analyzed Lebanese SARS-CoV-2 strains.

SARS-CoV-2 Strain	AA Substitution	Prevalence in Lebanese Strains	Prevalence between the Strains Reported on GISAID ^1^	First/Only Report According to GISAID’s CoVsurver	Number of Countries	Strain Collection Date
Spike protein						
LAU-R186	spike F59Y	1	106	First	4	01-Aug-20
	spike G35S	1	6	First	2	
	spike N17S	1	797	First	8	
	spike R44S	1	2	First	2	
	spike Y38stop	1	1	Only	1	
LAU-R45	spike P39T	1	8	First	6	01-Jun-20
	spike S31Y	1	1	Only	1	
	spike V11D	1	1	Only	1	
LAU-R204	spike L10R	1	1	Only	1	01-July 20
	spike L5P	1	2	First	2	
LAU-R87	spike D40V	1	1	Only	1	01-Sep-20
LAU-R129	spike L611R	1	5	First	2	01-September-20
LAU-R151	spike P82T	1	3	First	2	01-September-20
LAU-R43	spike Y91stop	1	1	Only	1	01-Jun-20
	spike F92L	1	2	Last	2	
Nucleocapsid protein						
LAU-R145	N E367stop	1	3	First	3	01-Oct-20
LAU-R231	N L352stop	1	1	Only	1	01-Aug-20
LAU-R43	N L407M	1	1	Only	1	01-Jun-20
LAU-R153	N M411K	1	1	Only	1	01-September-20
LAU-R92	N S318W	1	1	Only	1	01-September-20
Envelope protein						
LAU-R110	E C44S	1	6	First	3	01-September-20
S6	E V62D	1	1	Only	1	09-Mar-20

^1^ Up to 18 April 2021.

## Data Availability

SARS-CoV-2 sequences are available on GISAID [40] and NCBI’s GenBank [41].

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
