# Peer review of "SARS-CoV-2 Variants in Lebanon: Evolution and Current Situation"

_biology, 2021, doi:10.3390/biology10060531_

Round 1

Reviewer 1 Report

This study analyzed 58 SARS-CoV-2 Lebanese strains sequenced, and found 22 uncommon mutations and an interesting combination of mutations in S protein.  The authors provide good background and reasonable conclusion supported by their results. However, I think this study was limited by the small number of sequences analyzed, which was also mentioned by authors. Here, I also would like to invite authors to do the following revision before its acceptation:

  1. Graphical Abstract:the resolution of the figure is very low to me, it would be great if you can improve it
  2. Line 60-61: missed the reference
  3. Line 82-83: This is likely due to a change in the conformation of the S protein, resulting in better binding with the host’s receptor protein. Please name out this receptor protein here (ACE2)
  4. Could you discuss why the D614G strain is most prevalent in Lebanon.
  5. According to GISAID’s CoVsurver, is it possible to narrow the sequence samples onlyin Lebanon, if the authors would like to investigate the SARS-CoV-2 variants in Lebanon like the title mentioned? Please discuss.

Author Response

Kindly find attached the reply to your comments. 

Reviewer 2 Report

The manuscript entitled "SARS-CoV-2 variants in Lebanon: Evolution and current situation". Title, abstract and overall rationale of work to some extent is good and novel. However, there are still some minor concerns, which needs to be addressed and needs substantial revision.

1) Author must increase the resolution of graphical abstract at least 300dpi. Figure 1 figure 2 and  figure 3 also need to increase resolution and figure 2 in the bottom table naming size should be increase.

2) I would suggest the authors to enhance your theoretical discussion and arrives your debate or argument.

3) Conclusion section also needs to write about the significance of work and future prospective of this work.

4) Authors have mentioned very little about COVID-19 in introduction. Without the start of fundamentals about the subject in any manuscript doesn’t provide understanding to all types of readers. Therefore, it should be reader friendly with a proper flow. Authors can use below mentioned references, which will help them in adding this paragraph in introduction and discussion section and can be cited.

doi:10.1080/07391102.2020.1802345.

5) A flowchart should be added to this research article to show the clear methodology and mechanism.

Author Response

(The authors gave the same response as above.)
